# The dental triage method at Rothschild Hospital during the first lockdown due to the COVID-19 pandemic

Yara Saade[1,2], Muriel de la Dure Molla[1,3,4], Benjamin P. J. Fournier[1,4,5], Stéphane Kerner[1,2,6], Pierre Colon[1,7,8], Maria Clotilde Carra[1,2,9], Philippe Bouchard[1,2,10]*

1 Service d'odontologie, Université Paris Cité, Paris, France, 2 Department of Periodontology, Service d'odontologie, AP-HP Rothschild Hospital, Paris, France, 3 INSERM UMR S1163 Bases Moléculaires et Physiopathologiques des Ostéochondrodysplasies, Institut Imagine, Necker, Paris, France, 4 Reference Center for Oral and Dental Rare Diseases (ORARES), AP-HP Rothschild Hospital, Paris, France, 5 Centre de Recherche des Cordeliers, INSERM UMRS 1138, Molecular Oral Pathophysiology, Université Paris Cité, Sorbonne Université, Paris, France, 6 Department of Periodontology, Loma Linda University School of Dentistry, Loma Linda, California, United States of America, 7 Department of Conservative Dentistry and Endodontics, Service d'odontologie, AP-HP Rothschild Hospital, Paris, France, 8 Laboratoire des Multimatériaux et Interface, Université Claude Bernard Lyon 1 UMR CNRS 5615, Lyon, France, 9 Population-Based Epidemiologic Cohorts Unit, INSERM UMS 011, Villejuif, France, 10 Institut des Maladies Musculo-Squelettiques, Orofacial Pathologies, Imaging and Biotherapies Laboratory URP2496, Montrouge, France

* philippe.bouchard.perio@gmail.com

**Data Availability Statement:** All relevant data are within the manuscript and its Supporting Information files.

## Abstract

### Objective

This study aims to (1) assess the efficacy of a face-to-face emergency protocol in children and adults and (2) measure the efficacies of prediagnosis at the triage level and clinical diagnosis at the emergency department level during the COVID-19 pandemic.

### Methods

A triage protocol was applied for patients at the entry of the Rothschild Hospital (AP-HP) between March 18th and May 11th, 2020. First, patients underwent a triage based on self-reported symptoms. If their condition was deemed urgent, they were oriented toward dental professionals, who performed an intraoral examination leading to a clinical diagnosis. Triage and diagnoses were categorized into four emergency groups: infectious, prosthetic, traumatic, and others. The agreement between triage and clinical diagnosis was tested ($\chi^2$ test). Positive predictive value (PPV), negative predictive value (NPV), sensitivity, and specificity for each diagnostic category were assessed to evaluate the performance and efficacy of the triage.

### Results

Out of 1562 dental visits, 1064 were included in this analysis. The most frequently reported symptoms by children at triage were pain (31.5%) and trauma (22%). Adults mainly complained of abscesses (45.1%) and pulpitis (20.5%). The most frequent clinical diagnoses

**Funding:** The author(s) received no specific funding for this work.

**Competing interests:** The authors have declared that no competing interests exist.

were abscesses (29.2%) and pulpitis (20.5%) among children and adults, respectively. Tooth extraction was the most frequent treatment modality. Systemic antibiotics were prescribed for 49.2% of patients. Regardless of the age class, the PPV was high for groups 1 to 3, ranging from 78.9% to 100%. The NPV was high in all groups, ranging from 68.8% to 99.1%.

## Conclusion

This study demonstrates that the triage implanted during the first COVID-19 lockdown was effective and is an appropriate tool for the referral of adults and children before clinical examination.

## Introduction

Severe acute respiratory syndrome (SARS-CoV-2), also known as COVID-19, is a member of the Coronaviridae family. This virus was first identified in Wuhan, China, between October and December 2019. The virus spread to more than 185 countries by March 2020, thus acquiring pandemic status; as of January 2023, there have been more than 656 million confirmed cases and approximately 6.6 million related deaths [1, 2]. The most common symptoms that have been identified are fever, tiredness, and dry cough. However, in approximately 5% of cases, more severe symptoms, such as breathing difficulties, chest pain and loss of speech and movement, require hospitalization and admission into intensive care units [1, 3, 4]. The first case of COVID-19 in the French territories was occurred on January 19th, 2020, and was identified by January 24th, 2020. Following the announcement of the pandemic status on March 11th, the French government required a national lockdown on March 17th with the hope of achieving a sharp decline in the ongoing spread, similar to the decline observed in China. These measures resulted in a 77% decrease in virus transmission and an easing of the restriction measures on May 11th, 2020 [5] (S1 Fig).

Rothschild Hospital is one of the 39 hospitals belonging to the Assistance Publique-Hôpitaux de Paris (AP-HP), the university hospital center operating in Paris and its surrounding areas [6]. Rothschild Hospital has a department of physical therapy, a geriatric department and a department of odontology. Planned admission of ambulatory patients in the department of odontology can be done either in an on-site office, online or by phone call. Unplanned admissions are performed in the dental emergency room. A full-time dental professional sorted the patients into priority groups according to their needs. Patients were sorted into one of the following categories according to the World Health Organization recommendations [7]:

- Emergency patients requiring immediate treatment

- Priority patients requiring rapid treatment (priority in the queue)

- Nonurgent patients not requiring emergency treatment (waiting in the queue)

During the lockdown, the national "Plan Blanc" ("*White Plan*") was implemented. Plan Blanc is a specific health emergency and crisis plan to organize the rapid and rational implementation of essential resources in the event of an influx of victims into a hospital establishment. The nonmedical staff of the dentistry department was then relocated to the geriatric department, where the flow of patients in the emergency room had considerably increased. In addition, any staff member suspected of having COVID-19 infection or with potential

comorbidities was banned from the hospital. As a result, the odontology crew was considerably reduced. In the meantime, the National Dental Council had decided to apply strict measures that forbade dental practitioners from opening their private practices and treating patients. Therefore, in case of emergency, the patients were referred to the hospital, where only emergency patients requiring immediate treatment were admitted. Thus, the department of odontology had to deal with a decrease in the number of dental staff as well as an increase in the number of emergency patients.

The dental service had to evolve rapidly in response to the pandemic. Governmental guidelines and preventive measures for the management of dental patients were taken according to the directives of the French Ministry of Health [8]. In addition, a crisis management team (CMT) was established. The CMT led by the head of the department of odontology included a dental representative of each disciplinary field, a representative of the nonmedical staff, and a resident representative. A single route and space for the management of COVID-19 patients was identified within the hospital. This included a separate patient pathway and a dedicated dental office. This CMT implemented a simple diagnosis sorting system to provide a high-quality core service with an admission rate adapted to the reduction of the hospital workforce. The objective at the entry of the hospital was to identify COVID patients, to assess the level of emergency and to provide a prediagnosis to manage patient flow. Due to the shortage of hospital staff, undergraduate students were asked, on a voluntary basis, to fill either nonmedical staff positions or to participate in triage at the entrance of the hospital. Additionally, the clinical departments in the hospital were merged into one single emergency department, which included the 4 following branches: pediatric dentistry, prosthetics, endodontics, and oral diseases/oral surgery, where oral surgeons and periodontists were merged.

The aim of this report is to (1) describe the experience of Rothschild dental staff during the first lockdown and (2) measure the efficacies of prediagnosis at the triage level and clinical diagnosis at the emergency department level.

## Materials and methods

This observational study was based on a retrospective analysis of a database that collected the entry information of 1651 dental emergency patients attending the Odontology Department of the Rothschild Hospital during the 6-week lockdown in France (March 17 to May 11, 2020).

The present study was conducted in accordance with the ethical standards of the institutional and national research committee and with the 1964 Helsinki Declaration and its later amendments or comparable ethical standards. The Human Investigation Committee (IRB) of the AP Sorbonne Université, site Hôpital Rothschild, 5 Rue Santerre 75012 Paris, approved this study (#TriDentRTH2022-20220610152256). Due to the retrospective nature of the study, informed consent was waived since the study data were collected in an anonymous manner and thus never linked to an individual.

When possible, initial screening took place remotely by triaging patients via telephone or teleconsultation. Switchboard operators were replaced by dental staff to best determine whether to postpone dental treatment. Otherwise, the screening for entry was done at the hospital door where the patient's medical and dental information were gathered on an Excel sheet using a PC laptop. Again, the aim was first to identify patients with COVID symptoms and then to sort the patients according to a dental diagnosis. The reader must keep in mind that at this stage of the pandemic in France, no antigenic test was commercially available. Only the RT-PCR test was available in a few hospitals in Paris. Rothschild Hospital could not perform PCR on site. Given the oral emergency, it was impossible to refer these patients to another hospital for testing. The body temperature of all the patients was measured with a tympanic

thermometer. According to the patient's self-report, the classic symptoms of COVID, namely, cough, fatigue, ageusia, and anosmia, were recorded. All patients suspected of having COVID were accompanied via a specific pathway to the dedicated COVID area, where they were treated for their oral emergency. They were then asked to be tested by PCR in another hospital.

On the basis of the patients' self-reported dental symptoms, triage based on a prediagnosis was performed by an undergraduate student under the supervision of a full-time dental professional working at Rothschild Hospital. Patients whose prediagnosis was deemed urgent were admitted to the hospital for treatment.

A standard oral examination including a panoramic radiograph was then performed by residents, registrars, or senior consultants. This clinical examination resulted in a diagnosis and tailored treatment. The clinical diagnosis as well as the treatment modality and the medical prescription were recorded according to the usual procedure in the department's emergency book. When patient records were available for individuals who already attended the Rothschild Hospital, medical and dental information was checked for potential comorbidities. The entry data and the emergency book data were merged in one Excel sheet for comparison (S2 Fig).

### Statistical analysis

For the analysis, triage and diagnoses were categorized into four emergency groups: inflammatory/infectious (group 1), prosthetic (group 2), trauma (group 3), and others (group 4). The distribution of continuous variables was described as the mean ± standard deviation. Categorical variables were quantified as percentages. *To* determine the performance and efficacy of the triage, the PPV, NPV, specificity, and sensitivity were assessed. The $\chi^2$ test was used to compare the prediagnosis at the triage level and the clinical diagnosis. The significance value was set as p value = 0.05. The analysis was performed using SPSS 25.0 (IBM).

## Results

Out of the 1562 visits at Rothschild Hospital, 1064 were included in the analysis since 14 patients could not be identified in the Excel file, 55 subjects not considered emergency patients were not accepted into the hospital for clinical examination, 127 patients visited the hospital for at least the second time, and 302 patients had incomplete data (Fig 1).

There were 52 visits in the first week; the number of weekly visits reached a peak at week 6 and then decreased in the seventh and eighth weeks, reaching 124 and 130 visits, respectively (S3 Fig). Table 1 indicates the characteristics of the sample. According to the policy of the department, patients younger than 17 years old were assigned to pediatric dentists. The pediatric care unit received 454 patients (43%) during the lockdown.

Fig 2A shows the detailed frequencies of children's chief complaints at the prediagnostic triage level, and the clinical diagnosis at the emergency level is given in Fig 2B.

Among children, the chief complaints at entry were pain (31.5%) and trauma (22%), and the most frequent diagnoses were abscesses (29.2%), trauma (18.9%) and dental caries (14.3%).

Among adults, abscesses (45.10%), pulpitis (20.50%) and pain (17.9%) were the most frequent prediagnoses (Fig 3A), while pulpitis accounted for 42.10% of the diagnoses (Fig 3B).

A majority of children (52.9%) and adults (58.5%) were admitted and treated within the hospital. S4 Fig indicates the treatments received by the patients following the clinical diagnosis. Regardless of the age category, extraction was the most frequent intervention, accounting for approximately one-fifth of the overall therapies.

As shown in S1 Table, among children, there was little difference between the prediagnosis and the diagnosis. Among adult patients (S2 Table), the distribution of the number of subjects

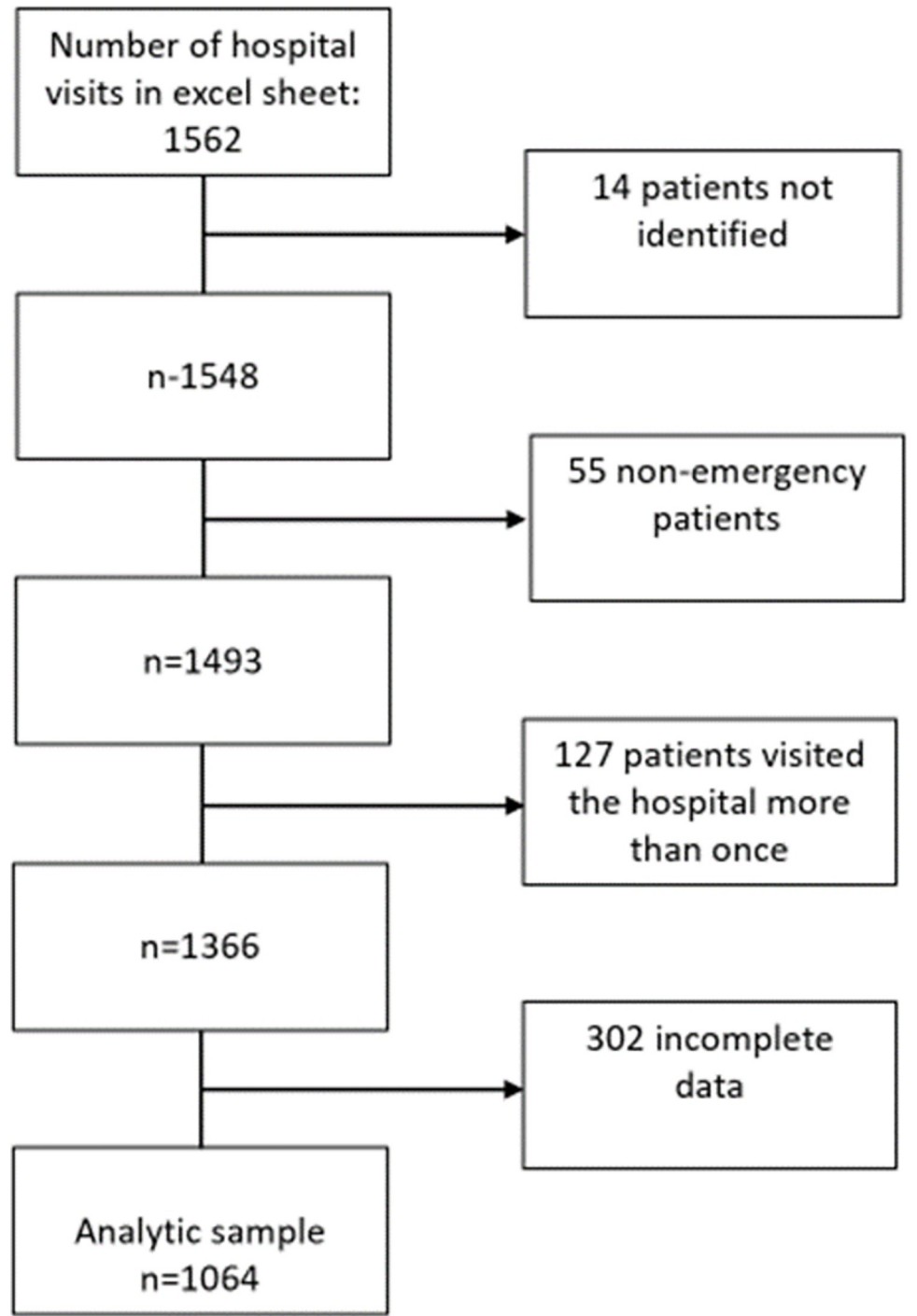

**Fig 1. Flow chart of patient selection.**

was homogenous in group 1, but there were some differences between the triage and the diagnosis for the other groups.

The PPV (i.e., the probability that the prediagnosis at the triage corresponds to the clinical diagnosis) and NPV (i.e., the probability that a false prediagnosis at the triage level corresponds to a wrong diagnosis) are shown in Tables 2 and 3 for adults and children, respectively.

**Table 1. Characteristics of the sample (n = 1064).**

|  | <17 years old 454 (43%) | ≥17 years old 610 (57%) |
|---|---|---|
| **Gender n (%)** |  |  |
| **Male** | 259 (57.0) | 325 (53.3) |
| **Female** | 195 (43.0) | 285 (46.7) |
| **Age mean ± SD** | 8 ± 3 | 44 ± 17 |
| **Temperature* mean ± SD** | 36.5 ± 1 | 36.5 ± 0.5 |
| **Tobacco smoking n (%)** |  |  |
| **Non-smoker** | 454 (100) | 448 (73.4) |
| **Former smoker** | 0 (0.0) | 22 (3.6) |
| **Current smoker** | 0 (0.0) | 140 (23.0) |
| **Diabetes n (%)** | 1 (0.2) | 24 (3.9) |
| **CVD n (%)** | 7 (1.5) | 51 (8.4) |
| **Respiratory disease n (%)** | 23 (5.1) | 19 (3.1) |
| **Auto-immune disease n (%)** | 0 (0.0) | 1 (0.2) |
| **Osteoporosis n (%)** | 0 (0.0) | 2 (0.3) |
| **Cancer n (%)** | 3 (0.7) | 13 (2.1) |

*n = 403 and 559 for < 17 years old and ≥ 17 years old, respectively.

Among the sample of adult patients admitted to Rothschild Hospital (Paris) during the COVID-19 lockdown (n = 610), the PPV for groups 1 to 3 was high, ranging from 78.9% to 89.3%. The NPV was high for all groups, ranging from 68.8% to 98.0%. The highest sensitivity value, which corresponded to the highest proportion of patients having a prediagnosis in accordance with the clinical diagnosis, was observed in group 2 for adults (99.8%). The specificity value of the prediagnosis was rather low in group 1 for adults (48.2%), and higher in the remaining groups ranging from 91.5 to 99.8%.

Among the sample of children patients admitted to Rothschild Hospital (Paris) during the COVID-19 lockdown (n = 454), the PPV for groups 1 to 3 was high, ranging from 95.0% to 100%. The NPV was high in all groups, ranging from 82.0% to 99.1%. The highest sensitivity value, which corresponded to the highest proportion of patients having a prediagnosis in accordance with the clinical diagnosis, was observed in group 3 for children (93.2%). The specificity value of the prediagnosis was high for children ranging from 87.2% to 100% across all groups.

## Discussion

The present study reports the experience of flexibility in the dental department of a public hospital in an industrialized country facing an epidemic. The outcomes show good concordance between the prediagnosis at the triage level and the clinical diagnosis. The efficacy of triage in assigning patients to adequate care is confirmed by the adequacy between the clinical diagnosis and treatment modalities.

The progression of patients' weekly visits is consistent with a study conducted in the UK mentioning an increase in clinical consultations from week 1 to 4 of the lockdown period, reaching a peak at weeks 5 and 6 [9]. It should be considered that at the time of data collection, no private health services were open. Thus, the results of our study must be seen from this perspective. It is possible that the patient profiles and chief complaints as well as the patient volume would have been different if the private practice had been open. The flow chart indicates that 127 patients visited the hospital more than once. Therefore, it can be assumed, as the

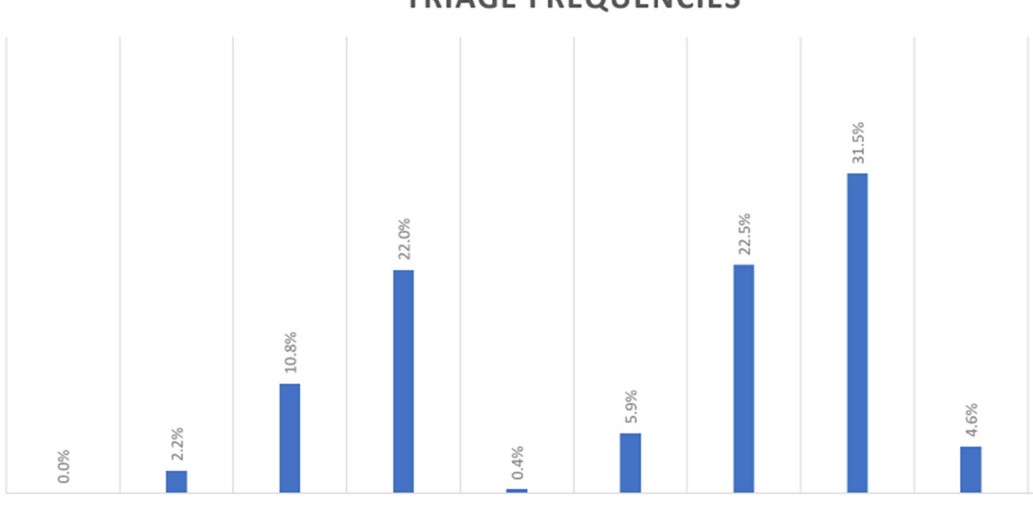

**Fig 2. Children.** (A) Prediagnosis at the triage level. (B) Diagnosis at the emergency level.

private dental offices were closed, that all the patients analyzed were consulting a professional for the first time because of the emergency that brought them to the hospital. It can also be assumed that the geographical recruitment of patients was in the neighborhood of the hospital because public transportation was restricted during the lockdown. However, we have no geographical information about the patients other than their personal address, which does not necessarily indicate their location during the lockdown.

The characteristics of the patients attending the hospital during the lockdown period were consistent with those of patients needing emergency dental treatment in France. A cross-sectional study conducted in Clermont-Ferrand Hospital in 2007 ascertained the profile of emergency patients. The main reason for consultation was pain, and male patients were slightly more prone to attend the hospital unit [10]. The methodology used in this report followed the

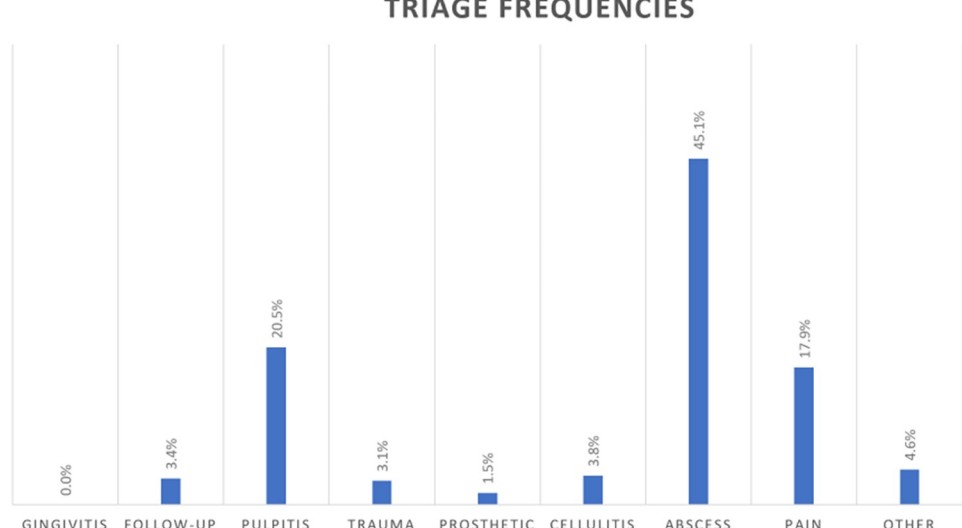

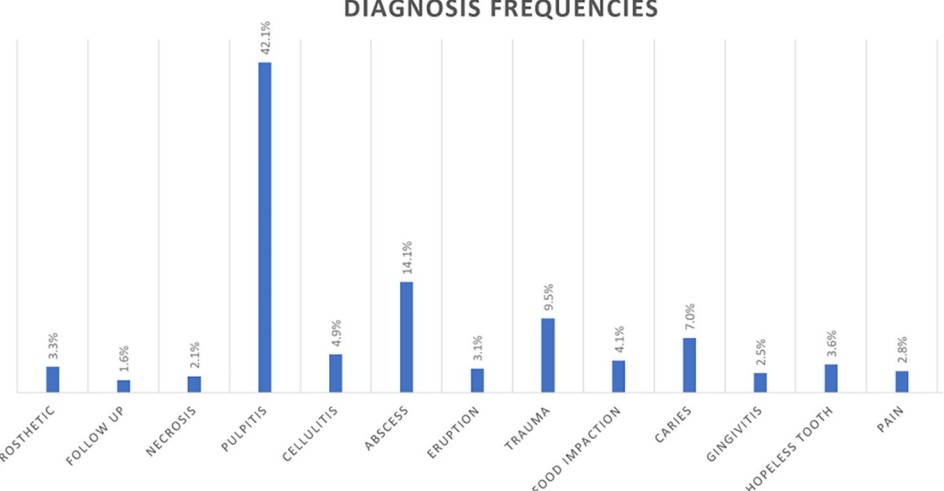

**Fig 3. Adults.** (A) Prediagnosis at the triage level. (B) Diagnosis at the emergency level.

guidelines spread in more than 16 countries encouraging patient triage through telephone as well as temperature screening at reception [11]. Other management modalities have also been described more precisely in a study from a Belgian team that evaluated the success rate of

**Table 2. Predictive values, sensitivity, and specificity of prediagnosis at the triage level versus clinical diagnosis in adults (%).**

|  | PPV | NPV | Specificity | Sensitivity | p value* |
|---|---|---|---|---|---|
| Group 1 Inflammatory/infectious | 89.3 | 68.8 | 48.2 | 95.2 | 0.0001 |
| Group 2 Prosthetic | 88.9 | 98.0 | 99.8 | 40.0 | 0.0001 |
| Group 3 Trauma | 78.9 | 92.6 | 99.3 | 25.4 | 0.0001 |
| Group 4 Others | 13.8 | 97.8 | 91.5 | 40.0 | 0.0001 |

PPV: positive predictive value; NPV negative predictive value.

*p values calculated with a CHI-2 test.

**Table 3. Predictive values, sensitivity, and specificity of prediagnosis at the triage level versus clinical diagnosis in children (%).**

|  | PPV | NPV | Specificity | Sensitivity | p value* |
|---|---|---|---|---|---|
| Group 1 Inflammatory/infectious | 95.0 | 82.0 | 87.2 | 92.7 | 0.0001 |
| Group 2 Prosthetic | 100.0 | 99.1 | 100.0 | 33.3 | 0.0001 |
| Group 3 Trauma | 96.0 | 98.0 | 98.9 | 93.2 | 0.0001 |
| Group 4 Others | 12.1 | 96.4 | 93.3 | 21.1 | 0.018 |

PPV: positive predictive value; NPV negative predictive value.

*p values calculated with a $\chi^2$ test.

remote emergency management regarding symptom relief and pain control over a 1-month period. The high success rate of remote treatment modality (71.8%) was rather encouraging and illustrates the possibility to differentiate definitive treatment for at least a month by means of advice and/or drug prescription. In addition, systematic follow-up at 1 week and 1 month proved beneficial not only for verification of the treatment efficacy but also for patient reassurance [12]. In Rothschild Hospital, teledentistry was used for the first time during the lockdown period, but no data are available to evaluate the efficacy; however, teledentistry has been shown to be a suitable option to increase access to health care services to patients during the COVID-19 pandemic in light of social distancing and lockdown measures [11, 12].

As expected, pain was the most frequent chief complaint with children at the triage level (31.5%). This percentage was dramatically reduced to 3.5% after the clinical examination. This outcome may indicate that patients tended to overestimate pain to be accepted into the hospital. One can also suggest that the chief complaint of pain at the triage level was allocated among the different diagnostic types. Interestingly, pain, as a triage category, was not as frequent in adults (17.9%). The discrepancy about pain as a chief complaint between children and adults at the triage level may reflect the anxiety of the parents when a child is affected.

Leaving aside pain, trauma (18.9%) and abscess (29.2%) dominated the clinical picture in children. However, the diagnosis of dental caries seems to be the most common diagnosis among child emergencies [11, 12]. This difference in findings may indicate that during confinement, children had movement restrictions, stayed and played at home and were not allowed to play outside, promoting the risk of domestic accidents. Furthermore, in children, pain and trauma were the most frequent chief complaints at entry, whereas abscesses, trauma and dental caries were the most frequent diagnoses. This difference can be explained by the fact that dental professionals at the triage level were not trained for pediatric diagnostics.

Pulpitis (42.1%) dominated the clinical picture in adults. This finding is in line with others showing that endodontic emergencies, with symptomatic irreversible pulpitis being the most common, are the most prevalent dental emergency visits [8, 13]. Notably, no periodontal diagnosis was performed at the triage level (0% periodontal emergency), whereas after clinical examination, 2.5% of adults and 0.2% of children were classified in the periodontal emergency group. This confirms, on the one hand, that periodontal emergencies are not frequent and, on the other hand, that patients are not educated to identify periodontal concerns.

The treatments of numerous children (47.1%) and adults (41.5%) were postponed after being diagnosed in the emergency department. This is consistent with the feeling among hospital staff that many "emergency requests" were false, corresponding to increased anxiety, especially with children's consultations. Indeed, it has been shown that the occurrence of anxiety and mental health problems has increased in all populations during the COVID-19 pandemic [14]. Dental extraction was the main treatment modality. This finding is consistent with another report [8]. Indeed, this treatment option was the most common since, as mentioned

in the Cochrane report on the reopening of dental service, it was highly recommended to avoid the use of aerosol-generating procedures such as the use of dental turbines [15]. Because it was not possible to adequately treat endodontic lesions with rotary instruments, the primary management of dental emergencies, especially in cases of COVID-19-positive patients, was by pharmacological methods, i.e., painkillers and/or antibiotics [16]. This can explain the overuse of these prescriptions in the present study.

Finally, the probability that the prediagnosis at triage corresponds to the clinical diagnosis (PPV) is high in adults and in children for 1, 2, and 3, whereas the PPV is poor for group 4, being 13.8% and 12.1% for adults and children, respectively. This can be explained by the fact that this group is highly heterogeneous and has a small sample size. A high level of heterogeneity was also observed in group 1. This heterogeneity negatively impacts the specificity of group 1 in adults (48.2%) but not in children (87.2%), whereas the diagnosis of group 1 is less heterogeneous than in adults (S1 and S2 Tables). In future research, group 1 should be split into 2 or 3 categories to improve the specificity of each of them. The sensitivity value was rather low across groups except for trauma group 3 in children (93.2%). The fact that inflammatory/infectious group 1 shows high sensitivity is mainly due to the imbalance in sample size. However, one must keep in mind that the goal of triage is to have good specificity and PPV rather than good sensitivity; taken together, the present results indicate that the implemented protocol during the COVID-19 lockdown was effective in establishing an appropriate patient orientation prior to clinical examination and treatment implementation.

Our report has several strengths. Very few data are available on triage during the COVID-19 pandemic [17]. To the best of our knowledge, the present study is the first study to describe the implementation and quality of an in-person (i.e., nonremote) triage during the COVID-19 lockdown. Notably, dentists involved in the triage had different levels of clinical and professional experience. Despite the lack of calibration of practitioners, a good quality of triage was found. Furthermore, we report triage not only for adults but also for children. The sample size (n = 1064) was sufficiently large to make inferences from the data. However, the retrospective and monocentric design of the present report is a major limitation, as well as the heterogeneity of the different diagnosis categories. Moreover, since the COVID-19 infection status of the patients was not tested the day of admission, it is not possible to formally conclude on the safety of the present strategy. The risk of cross-contamination cannot be discounted since treating positive and asymptomatic patients remained possible. However, no case of professional contamination was detected among the dental staff. Finally, unfortunately, no data were available to evaluate the effectiveness of telephone-based triage during the lockdown.

## Conclusion

The proposed in-person triage allowed for efficient management of dental services in a pandemic situation. The triage protocol implemented herein appears to be an appropriate tool for (1) referring adults and children before clinical examination to the appropriate specialty service and (2) reducing the time required for the therapeutic decision-making process.

## Supporting information

**S1 Fig. Timeline of covid-19 evolution and lockdown restrictions in France (Based on Institut Pasteur's reports reference).**
(TIF)

**S2 Fig. Flowchart of data collection.**
(TIF)

**S3 Fig. Weekly patients' hospital visits during the lockdown period.**
(TIF)

**S4 Fig.** Treatment modalities for children (A) and adults (B).
(TIF)

**S1 Table. Distribution of the number of subjects among the 4 groups analyzed according to triage or diagnosis and their difference in children.** Details of the different types of diagnoses are given for each category.
(PDF)

**S2 Table. Distribution of the number of subjects among the 4 groups analyzed according to triage or diagnosis and their difference in adults.** Details of the different types of diagnoses are given for each category.
(PDF)

## Acknowledgments

The authors thank the AP-HP and express their deepest respects to the medical and non-medical staff of the Rothschild Hospital who did not fail during this difficult period of the pandemic.

## Author Contributions

**Conceptualization:** Maria Clotilde Carra, Philippe Bouchard.

**Data curation:** Yara Saade.

**Investigation:** Muriel de la Dure Molla, Benjamin P. J. Fournier, Stéphane Kerner, Pierre Colon, Maria Clotilde Carra.

**Methodology:** Maria Clotilde Carra, Philippe Bouchard.

**Project administration:** Philippe Bouchard.

**Supervision:** Maria Clotilde Carra, Philippe Bouchard.

**Validation:** Maria Clotilde Carra, Philippe Bouchard.

**Visualization:** Philippe Bouchard.

**Writing – original draft:** Yara Saade.

**Writing – review & editing:** Philippe Bouchard.

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
