## [Decision Letter · Decision Letter 0]

12 Dec 2022

PONE-D-22-22941The dental triage method of the Rothschild Hospital during the first lockdown due to COVID-19 PandemicPLOS ONE

Dear Dr. Bouchard,

Cher Philippe,

Thank you for submitting your manuscript to PLOS ONE. After careful consideration, we feel that it has merit but does not fully meet PLOS ONE’s publication criteria as it currently stands. Therefore, we invite you to submit a revised version of the manuscript that addresses the points raised during the review process.

Please have the manuscript revised by a native English speaker before submitting corrections.

We look forward to receiving your revised manuscript.

Kind regards,

Inge Roggen, M.D., Ph.D.

Academic Editor

PLOS ONE

Journal Requirements:

Reviewers' comments:

Reviewer's Responses to Questions

**Comments to the Author**

1. Is the manuscript technically sound, and do the data support the conclusions?

Reviewer #1: Partly

Reviewer #2: Yes

2. Has the statistical analysis been performed appropriately and rigorously? 

Reviewer #1: Yes

Reviewer #2: I Don't Know

3. Have the authors made all data underlying the findings in their manuscript fully available?

Reviewer #1: Yes

Reviewer #2: Yes

4. Is the manuscript presented in an intelligible fashion and written in standard English?

Reviewer #1: Yes

Reviewer #2: Yes

5. Review Comments to the Author

Reviewer #1: Congratulations to the Authors for an extensive data curation.

However a few comments.

The authors mention that at the triage the patients symptoms in their own words were recorded. Here a sample triage form could be attached for better understanding of the readers.

The patients mentioned about the clinical diagnosis made. But fail to elaborate how the Dental healthcare workers performed it. It should reflect in the methodology.

Also another important point to be mentioned could be that the authors could mention about from where the patients came to the hospital, and which visit it was first or second. This could be potentially discussed in the discussion. As during lockdown (First stage) the travel means were also at a stand still.

Another important issue that the authors need to address in the manuscript that what happened to the patients giving COVID like symptoms. Were they tested and if so what were the percentage that were tested positive. this is a very important finding and would help in increasing the strength of triage being more effective. Also which test was done to confirm either a Rapid antigen of RT-PCR? This is an import aspect to be mentioned while putting up a paper on Triage. The authors can refer to a paper by Shinde et al., 2021

Int J Environ Res Public Health. 2021 Jul 8;18(14):7314. doi: 10.3390/ijerph18147314 and

Yu et al., J Endod 2020 Jun;46(6):730-735. doi: 10.1016/j.joen.2020.04.001. Epub 2020 Apr 10

Reviewer #2: This is a well conducted study with a good sample size. The following comments should be addressed,

I recommend language revision by a native speaker. There are some writing errors, some of which mentioned below.

Abstract:

There is no reference to COVID-19 in the objective of the study.

Methods: The settings of the study should be stated.

Introduction:

Line 51: please update the statistics.

Line 52-54: The sentence lacks a verb.

Line 63: sorts

Line 34: please include (PPV) and (NPV). After this, you can use just the abbreviations. Please check throughout the whole text.

Line 73: suspected for COVID-19 infection

Line 81: please cite the resource as a reference.

Line 82: led

Line 87: COVID-19. Please check throughout the whole text.

Materials and Methods:

Line 97: was

Line 100: please use “the present study” instead of “this study” when referring to your study.

All variables extracted from the patients’ files and records should be introduced and explained here. Line 123-125 should be added after that, and not under statistical analysis subheading.

Results:

Each table and figure should be self-standing. So, please add “among a sample of patients admitted to Rothschild hospital (Paris) during COVID-19 lockdown (n=…)” where appropriate.

Lines 161-162: please specify that this is for children.

Table 2: 0.0001 in p value column. The same for table 3. You can even delete this column and say in the footnote that all p values were <0.001.

Line 190-192: please revise the sentence, and preferably omit the repetitive “corresponding” term.

Please revise the format of the S1 and S2 tables, as there is no line in them.

Discussion:

No need to mention the figures and tables in the discussion.

Line 205: …were consistent with those…

Line 225, 232: dominated.

6. PLOS authors have the option to publish the peer review history of their article (what does this mean?). If published, this will include your full peer review and any attached files.

Reviewer #1: No

Reviewer #2: No

---

## [Author Response · Author response to Decision Letter 0]

19 Jan 2023

PONE-D-22-22941

The dental triage method of the Rothschild Hospital during the first lockdown due to COVID-19 Pandemic

Comments to the Author

Reviewer #1

Congratulations to the Authors for an extensive data curation.

Thank you for the kind words that are highly appreciated.

However a few comments.

Q. The authors mention that at the triage the patient’s symptoms in their own words were recorded. Here a sample triage form could be attached for better understanding of the readers.

A. Thank you for this comment. Unfortunately, no triage form is available. Given the emergency of the situation together with the unknowns, we tried to take care of the most urgent matters. Patients’ characteristics and symptoms were recorded on an XLS spreadsheet with a laptop at the door of the hospital before admission. This is indicated p.6 l.127-129 of the manuscript as follow: “the screening for entry was done at the hospital door where the patient’s medical and dental information were gathered on an Excel sheet using a PC laptop.”

Q. The patients mentioned about the clinical diagnosis made. But fail to elaborate how the Dental healthcare workers performed it. It should reflect in the methodology.

A. Thank you for this remark. Change has been made p.6 l143-145 as follow: “A standard oral examination including a panoramic radiograph was then performed by residents, registrars, or senior consultants. This clinical examination resulted in a diagnosis and tailored treatment.”

Q. Also another important point to be mentioned could be that the authors could mention about from where the patients came to the hospital, and which visit it was first or second. This could be potentially discussed in the discussion. As during lockdown (First stage) the travel means were also at a standstill.

A. Thank you for this thorough remark. We tried to answer the question in the introduction section as follow (p.4; l.86-90): “In the meantime, the National Dental Council had decided to apply strict measures that forbade dental practitioners from opening their private practices and treating patients. Therefore, in case of emergency, the patients were referred to the hospital, where only emergency patients requiring immediate treatment were admitted.”

Further, the Flow Chart (Fig 1) indicates that 127 patients visited the hospital more than once. Consequently, one can supposes, as the private dental practices were closed, that the analyzed set of patients visited the dentist for the first time. Regarding the geographical recruitment of the patients, we do not have this information. 

As suggested by the reviewer, in order to clarify this important query, the following has been added in the flow of the manuscript (p.11-12; l 247-254): “The flow chart indicates that 127 patients visited the hospital more than once. Therefore, it can be assumed, as the private dental offices were closed, that all the patients analyzed were consulting a professional for the first time because of the emergency that brought them to the hospital. It can also be assumed that the geographical recruitment of patients was in the neighborhood of the hospital because public transportations were not available during the lockdown. However, we have no geographical information about the patients other than their personal address, which does not necessarily indicate their location during the lockdown.”

Q. Another important issue that the authors need to address in the manuscript that what happened to the patients giving COVID like symptoms. Were they tested and if so what were the percentage that were tested positive. this is a very important finding and would help in increasing the strength of triage being more effective. Also which test was done to confirm either a Rapid antigen of RT-PCR? This is an import aspect to be mentioned while putting up a paper on Triage. The authors can refer to a paper by Shinde et al., 2021

Int J Environ Res Public Health. 2021 Jul 8;18(14):7314. doi: 10.3390/ijerph18147314 and

Yu et al., J Endod 2020 Jun;46(6):730-735. doi: 10.1016/j.joen.2020.04.001. Epub 2020 Apr 10

A. At that stage of the pandemic in France, no antigenic test was commercially available. Only the RT-PCR test was available in a few hospitals in Paris. The Rothschild Hospital could not perform PCR on site. Given the oral emergency, it was impossible to refer these patients to another hospital for testing. Moreover, this was not the purpose of their visit, which was most often motivated by pain that had to be treated immediately. The patients with COVID-like symptoms were treated in a particular area as mentioned in the materials and methods section l.136-138, “All patients suspected of having COVID were accompanied via a specific pathway to the dedicated COVID area, where they were treated for their oral emergency.” Patients were then advised to get tested in another hospital but we do not have the number of positive patients.

As suggested by the reviewer, in order to clarify this important query, the following has been added in the flow of the manuscript (p.5; l. 130-134): “The reader must keep in mind that at this stage of the pandemic in France, no antigenic test was commercially available. Only the RT‒PCR test was available in a few hospitals in Paris. Rothschild Hospital could not perform PCR on site. Given the oral emergency, it was impossible to refer these patients to another hospital for testing.” Further at the end of the paragraph (;136-138): “All patients suspected of having COVID were accompanied via a specific pathway to the dedicated COVID area, where they were treated for their oral emergency. They were then asked to be tested by PCR in another hospital.”

Reviewer #2: This is a well conducted study with a good sample size. The following comments should be addressed,

Thank you for the encouraging report.

Q. I recommend language revision by a native speaker. 

A. Following the reviewer’s suggestion, the manuscript has been reviewed by a professional editing service. The editing certificate is provided. 

There are some writing errors, some of which mentioned below

Abstract: 

Q. There is no reference to COVID-19 in the objective of the study.

A. Thank you for this important remark. The following sentence has been updated: “This study aims to (1) assess the efficacy of a face-to-face emergency protocol in children and adults and (2) measure the adequacy of pre-diagnosis at the triage level and clinical diagnosis at the emergency department level during the COVID-19 pandemic.”

Q. Methods: The settings of the study should be stated.

A. The settings has been added as follow: “A triage protocol was applied for patients at the entry of the Rothschild Hospital (AP-HP) between March 18th and May 11th, 2020.” 

Introduction: 

Q. Line 51: please update the statistics.

A. The sentence is now: “The virus spread to more than 185 countries by March 2020, thus acquiring pandemic status; as of January 2023, there have been more than 656 million confirmed cases and approximately 6.6 million related deaths [1,2].” (L.57-59)

Q. Line 52-54: The sentence lacks a verb.

Line 63: sorts

Line 34: please include (PPV) and (NPV). After this, you can use just the abbreviations. Please check throughout the whole text.

Line 73: suspected for COVID-19 infection

Line 81: please cite the resource as a reference.

Line 82: led

Line 87: COVID-19. Please check throughout the whole text.

A. All changes have been made accordingly and are in red in the manuscript.

Materials and Methods: 

Q. Line 97: was

Line 100: please use “the present study” instead of “this study” when referring to your 

All variables extracted from the patients’ files and records should be introduced and explained here. Line 123-125 should be added after that, and not under statistical analysis subheading.

A. All changes have been made accordingly and are in red in the manuscript.

Results: 

Q. Each table and figure should be self-standing. So, please add “among a sample of patients admitted to Rothschild hospital (Paris) during COVID-19 lockdown (n=…)” where appropriate.

A. Done

Q. Lines 161-162: please specify that this is for children.

A. Done

Q. Table 2: 0.0001 in p value column. The same for table 3. You can even delete this column and say in the footnote that all p values were <0.001. 

A. We would like to keep this column the column because one p-value has been estimated at p <0.018

Q. Line 190-192: please revise the sentence, and preferably omit the repetitive “corresponding” term.

Please revise the format of the S1 and S2 tables, as there is no line in them.

A. All changes have been made accordingly

Discussion: 

Q. No need to mention the figures and tables in the discussion.

Line 205: …were consistent with those…

Line 225, 232: dominated.

A. All changes have been made accordingly

---

## [Editor Report · Decision Letter 1]

23 Jan 2023

The dental triage method at Rothschild Hospital during the first lockdown due to the COVID-19 pandemic

PONE-D-22-22941R1

Dear Dr. Bouchard,

Cher Philippe,

We’re pleased to inform you that your manuscript has been judged scientifically suitable for publication and will be formally accepted for publication once it meets all outstanding technical requirements.

Kind regards,

Inge Roggen, M.D., Ph.D.

Academic Editor

PLOS ONE
---

## [Editor Report · Acceptance letter]

27 Jan 2023

PONE-D-22-22941R1 

The dental triage method at Rothschild Hospital during the first lockdown due to the COVID-19 pandemic 

Dear Dr. Bouchard:

I'm pleased to inform you that your manuscript has been deemed suitable for publication in PLOS ONE. Congratulations! Your manuscript is now with our production department. 

Kind regards, 

on behalf of

Dr. Inge Roggen 

Academic Editor

PLOS ONE